# Sociodemographic Characteristics and Interests of FeverApp Users

**DOI:** 10.3390/ijerph18063121

**Published:** 2021-03-18

**Authors:** Silke Schwarz, David D. Martin, Arndt Büssing, Olga Kulikova, Hanno Krafft, Moritz Gwiasda, Sara Hamideh Kerdar, Ingo Fingerhut, Ekkehart Jenetzky

**Affiliations:** 1Faculty of Health, School of Medicine, Witten/Herdecke University, 58448 Witten, Germany; Silke.Schwarz@uni-wh.de (S.S.); David.Martin@uni-wh.de (D.D.M.); Arndt.Buessing@uni-wh.de (A.B.); Olga.Kulikova@uni-wh.de (O.K.); Hanno.Krafft@uni-wh.de (H.K.); Moritz.Gwiasda@uni-wh.de (M.G.); Sara.HamidehKerdar@uni-wh.de (S.H.K.); 2Department of Pediatrics, Eberhard-Karls University Tübingen, 72076 Tübingen, Germany; 3Practice Kleiner Piks, 44869 Bochum, Germany; fingerhut@kleiner-piks.de; 4Department of Child and Adolescent Psychiatry and Psychotherapy, University Medical Center of the Johannes-Gutenberg-University, 55131 Mainz, Germany

**Keywords:** fever, FeverApp, ecological momentary assessment, user behavior, sociodemographic characteristics, registry, guidelines, feasibility, usability

## Abstract

The FeverApp Registry is a model registry focusing on pediatric fever using a mobile app to collect data and present recommendations. The recorded interactions can clarify the relationship between user documentation and user information. This initial evaluation regarding features of participants and usage intensity of educational video, information library, and documentation of fever events covers the runtime of FeverApp for the first 14 months. Of the 1592 users, the educational opening video was viewed by 41.5%, the Info Library was viewed by 37.5%, and fever events were documented by 55.5%. In the current sample, the role of a mother (*p* < 0.0090), having a higher level of education (*p* = 0.0013), or being registered at an earlier date appear to be cues to take note of the training video, Info Library, and to document. The FeverApp was used slightly less by people with a lower level of education or who had a migration background, but at the current stage of recruitment no conclusion can be made. The user analyses presented here are plausible and should be verified with further dissemination of the registry. Ecological momentary assessment is used more than the information option, in line with the task of a registry. Data collection via app seems feasible.

## 1. Introduction

Mobile health applications (apps) are now widespread, although their use in Germany is still relatively low [1]. The Digital Health Care Act (DVG), which came into force in 2019, entitles approximately 73 million insured individuals in the statutory health insurance system to be provided with digital health applications (DiGA), which can be prescribed by physicians as an “app on prescription” and reimbursed by the health insurance fund. The introduction of this technology and accompanying structures creates challenges for patients, providers, and the industry. At the same time, it is becoming clear how large the influence of mobile technologies can be if they are used appropriately [2].

A child having a fever is one of the most frequent reasons for parents to consult a pediatrician [3,4]. Although a fever is known as a useful body reaction to fight the underlying pathogens [5], many parents feel insecure and anxious when dealing with a child with a fever. Lack of knowledge and wrong perceptions about fevers could be potential causes of this anxiety [6]. It has been recognized that parent anxiety could lead to mismanagement of fever [4], such as overuse of antipyretics [6,7,8] which could lead to unintentional poisoning [9,10]. Accordingly, researchers recommend not only clear communication between pediatricians and parents [11], but also educating parents about fevers and its benefits as well as about the warning signs that should induce seeking help [12,13,14,15]. Several studies have examined the influence of different manners of education regarding fever and found a positive outcome of attaining better knowledge and management of fevers [16]. However, in a scoping review, Arias et al. [17] concluded that both parents and healthcare personnel should be addressed by a “more standardized educational platform”.

In 2019, six model proposals to create patient-related registries to address important subjects in health service research were selected and funded by the Federal Ministry of Education and Research (BMBF) in Germany. The six model registries focus on different subjects [18,19]. One of them, the FeverApp registry, aims to advance knowledge on fevers. The multiple objectives are: To achieve ecological momentary assessment (EMA) of data on febrile illnesses, to “assess guideline adherence”, to increase the knowledge about parental fever management, to improve parental skill, knowledge, and confidence in fever management, and consequently, to reduce use of medication and overuse of healthcare visits [20]. For this purpose and as a means of data collection for the registry, a smartphone application, FeverApp, was developed. A particular potential for the users of health apps lies in the education about relevant topics and the possibility of increasing their own health literacy in order to be able to cope better with the course of diseases [21,22].

The question posed by this study is the sociodemographic characteristics of current users of the FeverApp and what feature is of interest to them according to their usage. This provides a basis for understanding which target group uses the app and which groups of users do not have good access to it, as well as which features of the app are relevant to which groups of users. This will assist in not only the planning of further educational content, specifically position based on user behavior, but also similar health apps with multiple features such as, visualization, information, and documentation. As this kind of documentation feeds the central registry for research on fever events, interactions in the app are monitored and first analyses are presented. 

## 2. Materials and Methods 

The FeverApp enables a parent to document their child’s febrile illnesses in real time (Ecological Momentary Assessment), to learn about fevers, and how to safely treat it [20].

The submitted entries and interactions between different pages of the app are stored locally in the app within an open-source JavaScript database, PouchDB, which synchronizes it when online with Apache CouchDB. The latter is centrally located on the University of Witten/Herdecke’s servers and is transformed daily into a Mongo-DB. Several relational data tables (CSV format) are extracted on demand through structured query language scripts and are processed in SPSS V25 (IBM Corp., Armonk, NY, USA, Appendix A). These data represent the registry. A positive vote by the ethics committee of the University of Witten/Herdecke on pseudonymized data collection using an app was received (#139/2018), as well as a positive vote by our data protection service. 

From September 2019 to July 2020, the FeverApp was accessible only through selected pediatric and adolescent practices for the validation of data and improvement of the app’s usability. Physician clearance is required to analyze the validity of the data collected. The comparison between the data systematically collected by the pediatricians in the practice and the FeverApp registry data will be published separately. Since July 2020, access was facilitated for all interested practices. Each practice received a unique code that was shared with parents. Upon entering the practice code in the app, a family code was generated for each family user. Families had the option of so-called “family sharing”, which means that an individual family code could be shared within a family with any number of smartphone users. Each smartphone user creates a user profile during installation, whereby all users (roles) of a family (code) could create profiles of their observed children or view them together. For each child (profile), one or more series of entries (loops, i.e., a series of entries) could be created over the course of a fever event. In addition, the users receive information on childhood fever via educational video (Figure 1a) and a detailed guideline orientated multimedia Info Library with 23 chapters (Figure 1b). After documentation of multiple entries (Figure 1c), users could have an overview of all entries within a fever event in the “graph view” section of the app (Figure 1d). To enhance user experience, the app has the option of night mode, in which the appearance of the app background changes from light to dark.

As shown in Figure 1b,d, at the bottom of the screen the user can choose to switch between the overview (i.e., Homepage) and Info Library, in which they have access to information on fevers (Figure 1b). Such navigations within the FeverApp, called interactions, are also recorded. Besides reporting FeverApp user characteristics, use of the documentation and information features is the focus of this first evaluation. For clarity, the presentation of user behavior distinguishes between those who did not use the documentation or the Info Library at all, slightly (one to two times), or intensively (more than twice). This is only possible by separately tracking the interactions made by the user. In this publication, the following were analyzed for the entire cohort and not only for selected users: The use of the opening video and guide information area as well as the documentation activity, i.e., the actual registry. The evaluation was carried out with IBM SPSS Statistics 25 (IBM Corp., Armonk, NY, USA) in the form of absolute and relative frequencies. The explorative p-value was determined with the help of the Chi^2^ test or Fishers exact test. We will report whether the information video intended to convey the core information was used, and by whom (Table 1). Furthermore, we compare the user characteristics with the use of information (Info Library, Table 2) and the documentation (registry entries, Table 3).

## 3. Results

### 3.1. Participants’ Description

For app use, 1451 families from 86 pediatric/adolescent practices registered by the time of this evaluation (31 October 2020). The largest practice recruited 45% of families (*N* = 649), the second largest 7% (*N* = 103), and 64 practices had just begun participation at the time of the evaluation and were feeding the registry with single-digit numbers of participating families. 

A total of 1592 users or installations of the app from 1451 families can be identified, not all of whom provided information on their sociodemographic data (marked with asterisks in the tables). Mothers were the most common family role at 83.4%, followed by fathers at 15.4%, and 1.2% consisted of other roles. The app was mostly used by one person in a family (91.4%). In 7.8%, two users entered the data and in 0.8%, more than two users entered the data. The platform used was slightly more often Android (57.3%) than iOS (42.7%).

Out of 1494 level of education entries, the majority of the users, 48%, had higher school education (‘Abitur’), i.e., general qualification for university entrance, 22.4% had ‘Fachhochschulreife’ (vocational diploma), 23.1% had ‘Mittlere Reife’ (intermediate school certificate), 5.5% had a ‘Hauptschulabschluss’ (basic school certificate), and 1.1% were without a school certificate. Of the users, 18.6% did not state their ethnic origin, and the remaining (*N* = 1296) were 89.0% German, 1.8% Turkish, and 1.4% Polish. The age of the users (*N* = 1503) was on average 35.5 ± 6.6 years with a range of 14–68 years (IQR = 31–39). Accordingly, 10 individuals would have been 14 years old. These are presumably misrepresentations, since the default setting indicates the year 2006, which must be actively corrected. Except for 89 users who did not indicate any age at all, 93.8% of the age data are available. 

We do not know the exact number of children per family since profiles created by users in the role of “mother” (*N* = 1136) were used as a substitute for the number of children. These were categorized as “users with one child” (61.2%), “with two children” (31.5%), or “with more than two children” (7.3%). Thus, on average, 1.16 ± 0.47 individuals entered data per family and created profiles of 1.38 ± 0.72 [range: 0–6] children. 

There was almost no difference between partners and single users when the user did not document at all (45.5% vs. 46%) or when they created at least some entries (53% vs. 53%). Since not all partners register, we do not know exactly how many of the families are single parent families and have therefore restrained from this analysis at this point in time.

### 3.2. Description of Interactions

The 1592 users performed a total of 175,564 interactions and 9275 entry series (loops). The median is 58 interactions (IQR = 30–133; range: 8–1815). The corresponding distribution is left-sloping, i.e., most users performed fewer interactions. Of the 175,564 interactions, the majority (58.2%) were related to a page selection. In 23.8%, it was starting, restarting, or pausing the app and in 9.1%, it was selecting a child’s profile. Deactivating night mode (8.0%) was significantly more common than activating night mode (0.4%). 

Furthermore, the 9275 series of entries were made by a total of 55.5% of all users, whereby these can be differentiated into low (17.2%) and more intensive users (38.3%) of the documentation function. The Info Library was visited by 37.5% of all users, of whom 15.0% viewed only one to two pages and 22.5% viewed more than two pages. Those who did not document anything also did not use the Info Library, while those who documented often also looked at the Info Library (Chi^2^ value: 315; *p* < 0.001).

### 3.3. Use of the Educational Video

In the following, we will report whether the information video that was intended to convey the core information was used, and by whom (Table 1). The four-minute educational video started at a total of 733 times (0.4% of all interactions). Out of the 1592 participants, 41.5% (*N* = 660) watched the video, of which some users (*N* = 43; 2.7%) watched it multiple times (two to seven times). In terms of family position, 43.4% are mothers, 34.7% are fathers, and 16.7% are others. Mothers watched the input video most often (*p* = 0.009). Those who watched the video also used the Info Library more intensively (*p* = 0.010; Chi^2^ value = 9.1) and documented in the app (*p* < 0.0001; Chi^2^ value = 28.9) (Table 3). There were no differences in opening and viewing the educational video among the four selected age groups (<30, 30–34, 35–39, and ≥40 years), but individuals with a higher level of education (see Table 1 for definition) were more likely (*p* = 0.0013; Chi^2^ value = 21.8) to view the video. Users who viewed the opening video tended to use the Info Library (*p* = 0.0104; Chi^2^ value = 9.1). Users who watched the educational video were definitely more likely to document in the FeverApp (*p* < 0.0001; Chi^2^ value: 29.7). However, the longer the installation of the app, the more the features were used (*p* = 0.0002; Chi^2^ value = 22.4).

**Table 1 ijerph-18-03121-t001:** Viewing the opening video.

Variable		All Users	No User of the Info Library (0 Interactions)	Low-intensity Users (1–2 Interactions)	High Intensity Users (>2 Interactions)	*p*-Value (Chi^2^-Value)
			Absolute frequency N (Relative frequency %)			
Type of role	Mother	1244 (83.4)	705 (56.7)	502 (40.4)	37 (3.0)	**0.0090** (Exact test) ^#^
	Father	230 (15.4)	150 (65.2)	76 (33.0)	4 (1.7)	
	Others	18 (1.2)	15 (83.3)	2 (11.1)	1 (5.6)	
	Total	1492 * (100)	870 (58.3)	580 (38.9)	42 (2.8)	
Education status	Highest (‘Abitur’)	717 (48.0)	386 (53.8)	311 (43.3)	20 (2.8)	**0.0013** (21.8)
	High (‘Fachhoch-schulreife’)	334 (22.4)	190 (56.9)	131 (39.2)	13 (3.9)	
	Moderate (‘Mittlere Reife’)	345 (23.1)	202 (58.6)	134 (38.8)	9 (2.6)	
	Low (‘Hauptschulabschluss’ or no certificate)	98 (6.6)	76 (77.6)	21 (21.4)	1 (1.0)	
	Total	1494 * (100)	854 (57.2)	597 (40.0)	43 (2.9)	
Operating system	Android	874 (57.3)	521 (59.6)	333 (38.1)	20 (2.3)	0.2724 (2.60)
	iOS	651 (42.7)	368 (56.5)	261 (40.1)	22 (3.4)	
	Total	1525 * (100)	889 (58.3)	594 (39.0)	42 (2.7)	
User registration	September to December 2019	404 (25.4)	198 (49.0)	194 (48.0)	12 (3.0)	**0.0002** (22.4)
	January to May 2020	306 (19.2)	180 (58.5)	116 (37.9)	10 (3.3)	
	June to October 2020	882 (55.4)	554 (62.8)	307 (34.8)	21 (2.4)	
	Total number	1592 (100)	932 (58.3)	617 (38.8)	43 (2.7)	
Age-group	<30 years	221 (14.7)	134 (60.6)	85 (38.5)	2 (0.9)	0.4023 (6.19)
	30–34 years	486 (32.3)	275 (56.6)	196 (40.3)	15 (3.1)	
	35–39 years	460 (30.6)	256 (55.7)	190 (41.3)	14 (3.0)	
	≥40 years	336 (22.4)	202 (60.1)	122 (36.3)	12 (3.6)	
	Total number	1503* (100)	867 (57.7)	593 (39.5)	43 (2.9)	
Migration status	German	1154 (89.0)	688 (59.6)	429 (37.2)	37 (3.2)	0.2229 (Exact Test) ^#^
	Other nationalities	142 (11.0)	84 (59.2)	57 (40.1)	1 80.7)	
	Total number	1296 * (100)	772 (59.6)	486 (37.5)	38 (2.9)	
Number of children/family	One child	695 (61.2)	381 (54.8)	299 (43.0)	15 (2.2)	0.1461 (6.81)
	Two children	358 (31.5)	200 (55.9)	142 (39.7)	16 (4.5)	
	More than two children	83 (7.3)	40 (48.2)	39 (47.0)	4 (4.8)	
	Total number	1136 ^$^ (100)	621 (54.7)	480 (42.3)	35 (3.1)	

* Missing values due to missing information up to user number 1592; ^#^ Fisher’s exact test instead of Chi^2^, due to too small cell population; ^$^ Based on 1136 mothers with profile information, not all 1592 users, therefore smaller number of cases. Bold if exploratory *p*-value < 0.01.

### 3.4. Use of the Info Library

The FeverApp Info Library contains 23 information pages, of which the “Warning signs of fever” section was visited most often (484 times; 11.5%). The Warning Signs section, located directly after the Fever video in position two of the Info Library, provides parents with information about critical signs and conditions for which a medical presentation should be made. Other preferred areas of information included “1. What is fever” (9.4%) with information regarding fever definition, “3. Certificate for employers” (6.7%) with information about German regulations about sick leave when one’s child is sick, and the table of contents (6.4%), also located at the top of the Info Library. Frequency of measurement (5.8%), accompanying symptoms (5.1%), and information on “correct” fever measurement (4.7%) were also visited slightly more often than the other information pages.

The use of the Info Library (Table 2) was divided into three groups: 22.5% used it more than three times, 15.0% seldom used it (once or twice), and 62.5% did not use it at all. It was mainly the fathers who did not use the Info Library (*p* = 0.0002; Chi^2^ value = 22.4), as well as users who had only recently installed the app (*p* < 0.0001; Chi^2^ value = 110.1). Age and number of children had no influence on the use of Info Library. The maximum level of the child’s fever (*p* < 0.001) was related to the intensity of Info Library use (Figure 2).

**Table 2 ijerph-18-03121-t002:** Use of the Info Library depending on the status of parents, education, age, and children.

Variable		All Users	No User of the Info Library (0 Interactions)	Low-Intensity Users (1–2 Interactions)	High Intensity Users (>2 Interactions)	*p*-Value (Chi^2^-Value)
		Absolute frequency N (Relative frequency %)				
Type of role	Mother	1244 (83.4)	759 (61.0)	191 (15.4)	294 (23.6)	**0.0002** (22.2)
	Father	230 (15.4)	177 (77.0)	25 (10.9)	28 (12.1)	
	Others	18 (1.2)	11 (61.1)	3 (16.7)	4 (22.2)	
	Total	1492 * (100)	947 (63.5)	219 (14.7%)	326 (21.8)	
Education status	Highest (‘Abitur’)	717 (48.0)	432 (60.3)	107 (14.9)	178 (24.8)	0.6414 (4.26)
	High (‘Fachhoch-schulreife’)	334 (22.3)	209 (62.6)	54 (16.2)	71 (21.3)	
	Moderate (‘Mittlere Reife’)	345 (23.1)	211 (61.2)	54 (15.7)	80 (23.2)	
	Low (‘Hauptschulabschluss’ or no certificate)	98 (6.6)	62 (63.3)	19 (19.4)	17 (17.3)	
	Total	1494 * (100)	914 (61.2)	234 (15.6)	346 (23.2)	
Operating system	Android	874 (57.3)	527 (60.3)	135 (15.4)	212 (24.3)	0.2867 (2.50)
	iOS	651 (42.7)	416 (63.9)	98 (15.1)	137 (21.0)	
	Total	1525 * (100)	943 (61.8)	233 (15.3)	349 (22.9)	
User registration	2019	404 (25.4)	188 (46.5)	69 (28.9)	147 (36.4)	**<0.0001** (110.1)
	January to May 2020	306 (19.2)	162 (52.9)	59 (19.3)	85 (27.8)	
	June to October 2020	882 (55.4)	645 (73.1)	111 (12.6)	126 (14.3)	
	Total number	1592 (100)	995 (62.5)	239 (15.0)	358 (22.5)	
Age-group	<30 years	221 (14.7)	140 (63.3)	35 (15.8)	46 (20.8)	0.7276 (2.32)
	30–34 years	486 (32.3)	292 (60.1)	73 (15.0)	121 (24.9)	
	35–39 years	460 (30.6)	273 (59.3)	75 (16.3)	112 (24.3)	
	≥40 years	336 (22.4)	216 (64.3)	50 (14.9)	70 (20.8)	
	Total number	1503 * (100)	921 (61.3)	233 (15.5)	349 (23.2)	
Migration status	German	1154 (89.0)	732 (63.4)	180 (15.6)	242 (21.0)	0.3361 (2.18)
	Other nationalities	142 (11.0)	99 (69.7)	18 (12.7)	25 (17.6)	
	Total number	1296 * (100)	831 (64.1)	198 (15.3)	267 (20.6)	
Number of children/family	One child	695 (61.2)	421 (60.6)	111 (16.0)	163 (25.3)	0.7182 (2.10)
	Two children	358 (31.5)	207 (57.8)	54 (15.1)	97 (27.1)	
	More than two children	83 (7.3)	47 (56.5)	15 (18.1)	21 (25.3)	
	Total number	1136 ^$^ (100)	675 (59.4)	180 (15.8)	281 (24.7)	
Educational video	No video	932 (58.5)	605 (64.9)	142 (15.2)	185 (19.8)	0.0104 (9.13)
	Video watched	660 (41.5)	390 (59.1)	97 (14.7)	173 (26.2)	
	Total number	1592 (100)	995 (62.5)	239 (15.0)	358 (22.5)	

* Missing values due to missing information up to the user count of 1592; ^$^ Smaller number of cases because the base is 1136 mothers with profile information and not all 1592 users. Bold if exploratory *p*-value < 0.01.

### 3.5. Documentation Use

Users (Table 3) who saw the educational video or consulted the Info Library were also more likely to document. These were primarily mothers and users from the early period (2019) and without a migration background. Users from 2019 also had the longest observation interval, so were more likely to have a fever event and hence documentation. 

**Table 3 ijerph-18-03121-t003:** Use of documentation function depending on the status of parents, education, age, and children.

Variable		All Users	No Fever Events Documented So Far (0 Entries)	Low Users of the Documentation (1–2 Entry Series)	Intensive User of the Documentation (>2 Entry Series)	*p*-Value (Chi^2^-Value)
		Absolute frequency N (Relative frequency %)				
Type of role	Mother	1244 (83.4)	533 (42.8)	213 (17.1)	498 (40.0)	**<0.0001** (32.4)
	Father	230 (15.4)	132 (57.4)	41 (17.8)	57 (24.8)	
	Others	18 (1.2)	14 (77.8)	4 (22.2)	0 (0)	
	Total	1492 * (100)	679 (45.5)	258 (17.3)	555 (37.2)	
Education status	Highest (‘Abitur’)	717 (48.0)	304 (42.4)	116 (16.2)	297 (41.4)	0.0316 (13.8)
	High (‘Fachhoch-schulreife’)	334 (22.3)	132 (39.5)	60 (18.0)	142 (42.5)	
	Moderate (‘Mittlere Reife’)	345 (23.1)	149 (43.2)	63 (18.3)	133 (38.6)	
	Low (‘Hauptschulabschluss’ or no certificate)	98 (6.6)	53 (54.1)	22 (22.4)	23 (23.5)	
	Total	1494 * (100)	638 (42.7)	261 (17.5)	595 (39.8)	
Operating system	Android	874 (57.3)	363 (41.5)	145 (16.6)	366 (41.9)	0.0621 (5.56)
	iOS	651 (42.7)	295 (45.3)	122 (18.7)	234 (35.9)	
	Total	1525 * (100)	658 (43.1)	267 (17.5)	600 (39.4)	
User registration	2019	404 (25.4)	74 (18.3)	66 (16.3)	264 (65.3)	**<0.0001** (310.0)
	January to May 2020	306 (19.2)	90 (29.4)	50 (16.3)	166 (54.2)	
	June to October 2020	882 (55.4)	545 (61.8)	158 (17.9)	179 (20.3)	
	Total number	1592 (100)	709 (44.5)	274 (17.2)	609 (38.3)	
Age-group	<30 years	221 (14.7)	102 (46.2)	47 (21.3)	72 (32.6)	0.0088 (17.1)
	30–34 years	486 (32.3)	209 (43.0)	85 (17.5)	192 (39.5)	
	35–39 years	460 (30.6)	179 (38.9)	68 (14.8)	213 (46.3)	
	≥40 years	336 (22.4)	159 (47.3)	59 (17.6)	118 (35.1)	
	Total number	1503 * (100)	649 (43.2)	259 (17.2)	595 (39.6)	
Migration status	German	1154 (89.0)	545 (47.2)	198 (17.2)	411 (35.6)	0.0103 (9.16)
	Other nationalities	142 (11.0)	61 (43.0)	39 (27.5)	42 (29.5)	
	Total number	1296 * (100)	606 (46.8)	237 (18.3)	453 (35.0)	
Number of children/family	One child	695 (61.2)	299 (43.0)	129 (18.6)	267 (38.4)	0.1314 (7.09)
	Two children	358 (31.5)	140 (39.1)	54 (15.1)	164 (45.8)	
	More than two children	83 (7.3)	29 (34.9)	16 (19.3)	38 (45.8)	
	Total number	1136 (100) ^$^	469 (41.2)	199 (17.5)	469 (41.3)	
Educational video	No Video	932 (58.5)	458(49.1)	168 (18.0)	306 (35.6)	**<0.0001** (28.85)
	Video watched	660 (41.5)	251 (38.0)	106 (16.1)	303 (45.9)	
	Total number	1592 (100)	709 (44.5)	274 (17.2)	609 (38.3)	
Info Library use	No use	995 (62.5)	582 (58.5)	187 (18.8)	226 (22.7)	**<0.0001** (315.4)
	Low use	239 (15.0)	69 (9.7)	47 (17.2)	123 (20.2)	
	Intensive use	358 (22.5)	58 (16.2)	40 (11.2)	260 (72.6)	
	Total number	1592 (100)	709 (44.5)	274 (17.2)	609 (38.3)	

* Missing values due to missing information up to the user count of 1592; ^$^ Smaller number of cases because the base is 1136 mothers with profile information and not all 1592 users. Bold if exploratory *p*-value < 0.01.

## 4. Discussion

The FeverApp is a documentation tool for the FeverApp registry, which also provides parents with guideline information on the subject of fevers to increase parental knowledge and confidence. While other health apps with a focus on fevers emphasize on surveillance of specific illnesses in specific seasons, such as influenza [23,24], Dengue fever [25], or malaria [26], the FeverApp registry concentrates on pediatric fever as a symptom and a common reason of parental anxiety and overuse of antibiotics, antipyretics, and health services. To the best of our knowledge, there are very few research-based mobile applications that focus both on educating parents and include a diary-function to record fever episodes and the way parents manage fever. 

The main users of the FeverApp are mothers (83%) in their 30s (63%), and persons with a higher level of education. These findings support the fact that younger individuals with a higher level of education are more likely to use mobile health apps [27]. The proportion of mothers with nationalities other than German using the app (11%) is lower than the reported, with 18% of non-German mothers living in Germany in 2019 [28]. However, at the current stage of recruitment no conclusion can be made. In 93% of cases, one to two child profiles are created, which reflects the reality of the number of children per family in Germany in 2019, where 88% have one or two children [29]. 

As indicated by other studies [30], it is important to maintain a close relationship between users and practitioners in order to achieve the best possible results from health apps. While recruitment style is possible through online advertisement [23] or word of mouth [24], in the FeverApp registry we recruited parents through pediatric and adolescent practices. More than half of the registry data collected are from the last five months, indicating evolving recruitment dynamics of providing access to the app through interested practices. 

The quality of the app and its user experience [31] determine if the users continue using the app for its intended purposes and therefore require constant evaluation [32]. The FeverApp and the registry data quality are regularly assessed in three different ways: (a) A voluntary feedback function in the app, (b) active qualitative testing with selected users, and (c) analysis of user behavior, i.e., interactions, based on registered data. The last category is the focus of the current article by investigating the level of interaction between information and documentation in detail. What is unique about this form of study is that it analyzes user behavior (regarding documentation in the registry and information) in the app by observing interactions. This user testing looks at the complete collective, as opposed to the other two forms of user feedback, which could become a new standard for health apps.

Data collection via the FeverApp offers an advantage, where: (a) Data collected are pseudonymized, (b) all answers are voluntary, and (c) users are informed, by pediatricians and through the app, that their entries are used for the purpose of research. Previous studies indicate that once individuals know that the intention of data collection is for research, they do not mind sharing their information [33], despite concerns about secondary use of information, e.g., by other companies [27]. Despite the fact that each part of data entry is voluntary, the data are relatively complete. 

Based on a systematic review, Young et al. [16] suggested that the best way to educate parents about fever management is a multidimensional education using different means such as text, video, and verbal materials. As mentioned before, the FeverApp’s Info Library is a multimedia library, consisting of videos, images, and text (Figure 1b, in addition to the documentation tool in the app which has of itself an educative aspect. The aim of the educational fever video developed in collaboration with the German Association of Pediatric and Adolescent Doctors (Berufsverband der Kinder- und Jugendärzte, BVKJ e.V.) is to inform parents about the essential elements of recommendations of the BVKJ e.V. and the German Society of Pediatrics and Adolescent Medicine (Deutsche Gesellschaft für Kinder- und Jugendmedizin, DGKJ) [34]. So far, 41.5% of all users who installed the app took note of the short video at the beginning. Although the video can also be accessed later via the Info Library, it was not frequently used there. Considering earlier research proposing that although parents appear to prefer videos in comparison with other means of education [17], knowledge transfer remains a challenge even with informative videos. However, it could be that previous users already have, or believe they have, a basic knowledge that is sufficient for them, so that further immersion seems unnecessary. Mothers and those with higher levels of education were more likely to watch the video than to use the Info Library. Likewise, parents who installed the app in 2019 were more likely to have watched the educational video (51%, Table 1).

It was found that people who watched the educational video also consulted the Info Library more intensively (26% versus 20%). Similarly, parents were more likely to use the Info Library if they had installed the app in 2019 (36%), i.e., the longer they had the app, the more likely they were to use it. For instance, 29% of the parents used the Info Library the first half of 2020 compared to 14% in the second half of 2020. The same trend was observed for viewing the educational video. This may be related to both the length of time these parents were observed in the registry and the type of application they used. Fathers were the least likely to use the Info Library intensively (12%). Mothers in their 30s (25%) consulted the Info Library slightly more often than those who are older or younger, although this is not statistically significant. This raises the question as to whether the younger, usually less experienced, individuals might already be well-informed and/or whether they seek their information from other sources. They may also be less aware of their need for information. It is striking that the 46% of the users who initially viewed the video about fever compared to the 36% who did not, were also those with intensive documentation. Those users (73% vs. 21%, Table 3) also used the Info Library. Both are strongly associated with intensity of documentation of fever to the maximum level of the child’s fever in the app (Figure 2). This would indicate the importance of parents having sound technical information on fevers at hand. With a higher temperature, more information seems to be needed, for instance, regarding medication, warning signs, etc. The app is most heavily used by parents between the ages of 30 and 40. Individuals with higher educational levels and users of a cell phone with Android also documented more. The above-average use of the iPhone (43%) is striking, as the national percentage of iPhone users was 29% in September 2020 [35]. The distribution of educational level among the users was in accordance with that of 30- to 40-year-old parents in Germany. 

One strength of this study in comparison with other studies on the subject of pediatric fever [23] is the collection of basic demographic information of users, in which the trend of fever management can be further investigated based on age, gender, etc. The strength of this registry is the current achieved number of cases. With nationwide coverage, the use of FeverApp is increasing. The extent of missing data is low despite the fact that it is completely voluntary. Although the unsystematic distribution of missing data limits the significance of the study, it does provide a naturalistic registry. The validation of the data through parallel recording in several family pediatric centers is currently taking place and will be analyzed separately. 

After one year, the registry data collected with the FeverApp provides a good overview of basic user behavior. This is important both for the further development, distribution strategy, and for the interpretation of analyses of the registry data collected with the app. Furthermore, the manner in which to present information in health apps could be changed depending on the sociodemographic of its users. 

## 5. Conclusions

The FeverApp provides the possibility to collect real-time data, educate parents about fevers, and provide insights to researchers and pediatricians regarding management of fevers at home. This study of the app’s interactions offers additional information on the behavior of the app’s users. It was shown that recording changes between screens could be useful. The recording and analyses of interactions could be extended, for example, more detailed analyses concerning the timing of the navigation between information and documentation. The educational opening video was viewed by 41.5% of the 1592 users, who were correspondingly more likely to also document fever events (62.0%; *p* < 0.0001) or consult the Info Library (40.9%, *p* = 0.0104). The documentation function was used more than the information option, whether via content of Info Library or an integrated educational video, in line with the task of a registry. It seems feasible to have data collection as a registry by means of an electronic case report app.

Overall, plausible trends could be demonstrated. The observation of user behavior was an important measure to further develop the registry and FeverApp. Whether people with a higher educational status, who use the app more frequently, are conversely more insecure in dealing with fever in their child than people from other educational backgrounds seems questionable, rather, there seems to be clearly a stronger thirst for information. The data collected provide an initial basis for controlling bias in this app-based registry. Communicating guideline knowledge is challenging. This specially developed app can do this in a way that is accessible to many and that can be monitored and continually optimized. User behavior of different subgroups in the registry and the potential clinical impact of information through the app will be further evaluated in the coming years.

## Figures and Tables

**Figure 1 ijerph-18-03121-f001:**
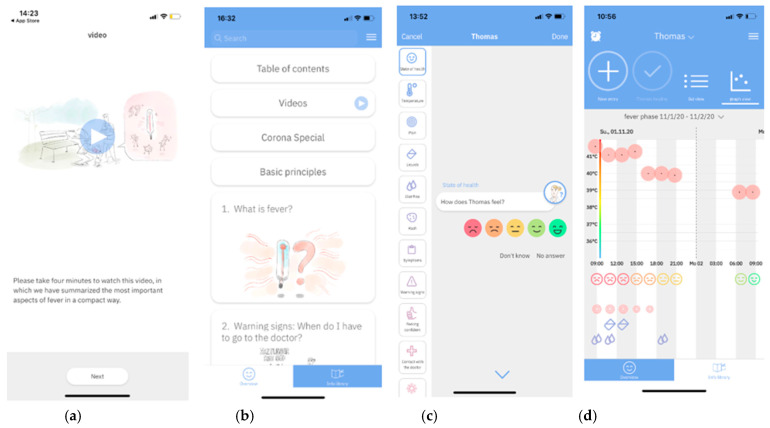
(**a**) Educational video; (**b**) Info Library-Menu; (**c**) Entry start; and (**d**) Graph view.

**Figure 2 ijerph-18-03121-f002:**
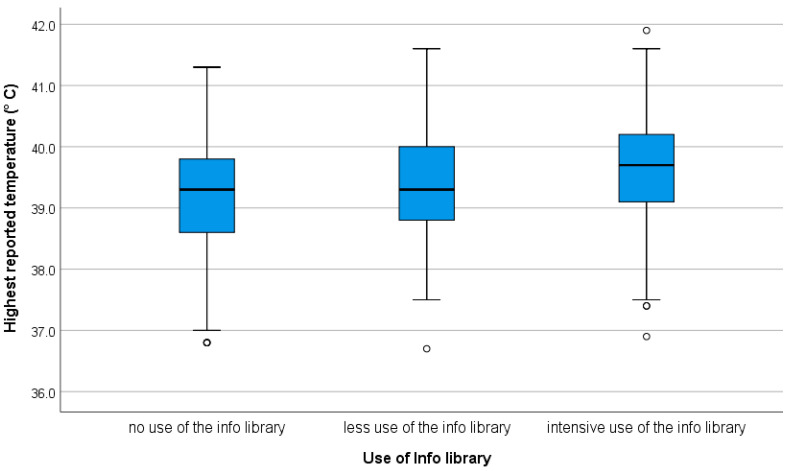
Use of the Info Library and highest reported temperature (circles are outliers).

## Data Availability

The data presented in this study are available in SPSS format as Appendix A.

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
