# Peer review of "Sociodemographic Characteristics and Interests of FeverApp Users"

_ijerph, 2021, doi:10.3390/ijerph18063121_

Round 1

Reviewer 1 Report

The authors have made substantial improvements in the text, and have addressed most of the concerns raised in the previous review.

Reviewer 2 Report

Dear authors,

After reading the revised manuscript, I almost understood what the Fever APP is.

However, as a reader, it was difficult to find value in the results of this study. If I were a developer or related person of this app, this research result could be very useful. But otherwise, I just could know who the Fever app is used for and for what purpose.

The tables show values that are statistically significant. But I couldn't know what is the practical worth. It may be useful data for developers who want to develop Fever APP.

In discussion, previous studies was cited and explanations were added. However, description was very abstractive, so I couldn't reach understanding the value of this research result.

I do not deny this app at all.
I hope more people will know this app and use it.
When I wrote this comment, I had a conflict in my heart.
I thought of my comments from the perspective of "as an academic paper".

Reviewer 3 Report

I think the shortening of the sections is good. The captions have been revised. Linguistically, the article has been revised. I like the changes to the tables. The abstract has also been revised. I have no further requests for changes. 

Reviewer 4 Report

The authors considered most of the the reviewer’s remarks appropriately. Two points remain that could be rethought. It should be left to the authors how to handle those points.

  1. The authors point several times to differences between mothers and fathers. The reviewer suggests adding an analysis differentiating between families having mother and father both as users on the one hand and families having either the mother or the father as user on the other hand. The reviewer hypothesizes that mothers and fathers behave similar if they are mainly responsible for their child. In the contrary, the mother dominates in the classical role model.

Response: Thank you for this interesting consideration. We tried to analyse your suggestion, but we did not find a major difference in viewing the video, using the Info library or document entries between families with both partners with installed app or single users with installed app. It may be that families with both partners having the app installed, documented more often than single users (42% vs. 35%). But there was hardly any difference between partners and single users, when they did not document at all (45.5.% vs. 46%) or when they created at least some entries (53% vs. 53%). Since not all partners register, we do not know exactly how many of the families are single parent-families and have therefore restrained from this analysis at this point in time.

Reviewer: The explanation could be added to the paper

  1. Line 25: “The FeverApp was used slightly less by people with low school qualifications or a migration background”. The reviewer asks himself whether this conclusion could be really drawn from the results. First, there might be a recruiting bias. Secondly, there might be a bias through the technical platform (iOS, Android). Thirdly, use could refer to the installation of the app or the frequency of app use. Etc. Especially the conclusion regarding the migration background seems questionable. Furthermore, the reference group should be mentioned here.

Response: We do not conclude the reason of this origin, we just recognize this difference, whether it is due to selection bias or anything else. The reviewer is correct, that there may by any kind of bias and therefore we analyse it. But to asses any difference with the nationwide reference data, we compare them with Destatis(cf.[29]) and can show the difference between the FeverApp collective and the complete population.

Reviewer: The reviewer recommends adding a remark that this observation does not meaning anything with regard to the influence of school qualification and migration background on the use of the app.

Author Response

This manuscript is a resubmission of an earlier submission. The following is a list of the peer review reports and author responses from that submission.

Round 1

Reviewer 1 Report

The paper presents a study with the sociodemographic characteristics of the users of the FeverApp, in use in Germany.

The study presents detailed analyses of the sociodemographic characteristics, breaking down the percentages of users and use features according to age, gender, ethnicity, education and other factors.

As a positive factor, the paper presents detailed descriptions of the statistical analyses and inferential statistical analyses performed, with justifications and rationale.  Raw data is also available in SPSS files with references in the paper, enabling replication of the analyses.

The authors present information about the use of video instructions and about the interactions users performed in the system.  However, I believe this could be better explored in the paper.  There is very little on the paper to help readers understand the context in which such actions take place.  It would be important to provide more details (maybe with visual cues) as to what steps are taken when those actions are performed.  This could be better discussed. 

Do authors foresee any potential investigations in terms of the usability of the system and features that could be deeper investigated from the evidence collected from the interactions?  That should be included in the discussion, and not have just shallow indications in the results.

Despite comparing the results with national statistics about the use of mobile phones and other sociodemographic characteristics of the German population, the paper does very little comparison of their results with other studies concerning the acceptance of mobile health apps.  Authors should perform a more thorough literature review of this type of study (including studies in this journal) and compare their results with studies performed in other countries).

Writing also needs revision.  There are many statements that do not read well in English.  Following I provide some examples of such excerpts.

Line 114: followed by 15.4% father a

Line 122: 1.4% Polish nationality

Line 143: a lot also looked at the Info library a lot

I believe the paper has potential to be accepted, since there is analysis of a significant body of data, with detailed statistical analysis.  However, the authors need to perform a more thorough literature review and to compare their results with other studies in the literature, as well as performing a careful proof-reading and copy-editing of their paper.

Reviewer 2 Report

I find the study they have carried out very interesting and very appropriate from the perspective from which they have approached it. However, I think it would be useful to expand the bibliography a little more with more recent studies, and to extend and present the conclusions in more detail.

Reviewer 3 Report

Dear Authors,

The article seems to be a kind of reports that summarizes the data rather than a article. I don't think it requires advanced statistical analysis. However, I think an article needs some kind of analysis. Doing a statistical test, however, the test is a test and is different from statistical analysis.
I don't think there is enough explanation overall.In particular, there is insufficient explanation of how this study can be useful.

Abstract
Overall,the expression in abstract is insufficient.

Line15
What is the "six model registry"?

Line 19 "As expected"
What did the authors expect?

Line 19 "middle-aged mothers represent the largest user-"
It doesn't show what kind of research it is. It seems to be written from the results, and the readers don't know what to interpret in this result.

Line 38
"The introduction of this technology creates challenges for the various stakeholders who are indirectly and directly affected."

What does that mean?

Line 44
"Although childhood fever is harmless in most cases, many parents feel insecure and anxious when dealing with it."
I don't deny it. However, the sentence is not objective and scientifically. What can be said to be almost harmless? Why can you say parents be said to be anxious?

Line46
Article [6] seems to be a little old. Are there newer articles?

Line 52
"The FeverApp registry is one of six model registries that have received public funding since 2019."
What is the "six model registry"?

Materials and methods

There is no explanation as to how the author collected the APP data.

Fig 1
If the authers show the contents of the app, it needs to explain the details of the screen a little more. For example, (c) "temperature graph is mentioned", but it does not look like a graph.

Line 99
The statistical test is explained, but there is no explanation as to what data was analyzed and how the analyze was performed.

Line 102
"User switching to documentation and information pages within the FeverApp is the focus of this first evaluation."
I don't understand the meaning of this expression.

Results
It may be significant because of the large number of samples. I think you should also show the effect size.

Discussion
It's more like a detailed explanation of the results than a discussion. I don't think there is enough explanation about what can be said from the results.

Reviewer 4 Report

The study describes the users of an app in great detail. Unfortunately, there is a lack of validation of the measurement by a second medical measurement.   I recommend that the tables be revised again, and that they be significantly shortened for the collective description.    Make it clearer what the added value of the app is.    As a result, it is a nice project. However, I would focus more on the benefits of the app and reduce the descriptions of the users. 

Reviewer 5 Report

The authors present an evaluation of a healthcare app about fever in childhood intended for use by parents. They describe sociodemographic patterns of the parents and the relationship between those patterns and the access of different app functions. The study shows that heavy users are better-educated mothers of middle age; the recording function of the app is more frequently used than the knowledge base about fever. The paper is well structured and easy to follow. Nevertheless, the paper would benefit from some major and minor revisions.

  1. The paper should be understandable without reading the study protocol published in reference 7. Therefore, the reviewer suggests adding some information about the data flow from the app to the registry that is repeatedly mentioned in the text.
  2. The discussion misses a comparison of the results with the state of the art. For example, it would be interesting whether the user and access profile is typical for healthcare apps or different. Furthermore, if information about apps related to families is available, it would be worthwhile to mention them.
  3. The authors point several times to differences between mothers and fathers. The reviewer suggests adding an analysis differentiating between families having mother and father both as users on the one hand and families having either the mother or the father as user on the other hand. The reviewer hypothesizes that mothers and fathers behave similar if they are mainly responsible for their child. In the contrary, the mother dominates in the classical role model.
  4. Line 15: „recorded“ is used two times. It could be replaced by “stored” one time.
  5. Line 20: Please clarify that “more likely” refers to the users who viewed the video. However, the percentages were calculated on the bases of all users.
  6. Line 25: “The FeverApp was used slightly less by people with low school qualifications or a migration background”. The reviewer asks himself whether this conclusion could be really drawn from the results. First, there might be a recruiting bias. Secondly, there might be a bias through the technical platform (iOS, Android). Thirdly, use could refer to the installation of the app or the frequency of app use. Etc. Especially the conclusion regarding the migration background seems questionable. Furthermore, the reference group should be mentioned here.
  7. Line 35: “low by international standards” Meaning unclear. Please rephrase.
  8. Line 48: The FeverApp is abruptly mentioned. There should be an extended introduction of the FeverApp.
  9. Line 64: Again, “recorded” is used two times.
  10. Line 65: The recruitment remains unclear. What was the difference between September 2019 and July 2020?
  11. Line 73: Please clarify the definition of a loop. Is loop a series of entries or is loop an entry?
  12. Figure 1: Parts c and d are inverted.
  13. Line 99: „the no-SQL data“ should be explained. This could be done in a separate paragraph briefly introducing the technical setting of the FeverApp.
  14. Line 115: “The app was mostly used by one person (86.7%).” This phrase is problematic. The app is usually used by one person at a single point in time. The reviewer assumes that the distribution of users to families is meant. This should be clarified by rewording the sentence. Furthermore, the percentages of 86.7%, 11.3%, and 2.0% from 1451 families do not add up to 1592 users. The reviewer calculated 1673 users. The percentages should be checked. Depending on the result, the percentages should be corrected or the meaning clarified.
  15. Line 118: “The majority of the users (N=1494)” Please clarify that 1494 is the total number of users with information on school education.
  16. Line 127: „Two thirds are between 30 and 40 years old.“ Redundant, because the IQR was mentioned before.
  17. Line 128: „Profiles created by users in the role of "mother" (N=1,136) were used as a surrogate for the number of children.“ Unclear, what surrogate means here.
  18. Line 133: Again, please clarify the definition of a loop.
  19. Line 134: The reviewer could not recalculate the mean number of “58 registered interactions” based on the presented figures.
  20. Line 134: Please define an interaction. If there is a difference between an “interaction” and a “registered interaction”, please mention. Please state, whether the recording of data is an interaction. Please clarify, whether a series of entries is an interaction or whether each single entry is an interaction.
  21. Line 137: Please explain the effect of the night mode on the app.
  22. How many users did not use the app at all?
  23. Line 146: „In the following, we will report whether the information video intended to convey the core information is used, and by whom (Table 1).“ Please shift his information to the method’s section.
  24. Line 148: „as an interaction“ - redundant
  25. Line 148: „This shows“ Reference of “this” unclear. Please rephrase.
  26. Line 151: “Mothers thus watched the input video most often (p=0.009).” See remark 3 of the reviewer.
  27. Line 155: How was „higher education level“ defined?
  28. Line 158: „However, the time since first using the app must also be taken into account here“ - unclear
  29. The layout of the tables 1 to 3 should be harmonized, e.g. by consistently presenting „Absolute frequency N (Relative frequency %)“ in a row, not in a cell as in table 1.
  30. Line 171: What is the „Certificate for employers“? How is this certificate supported by the app?
  31. Line 176: „It was mainly the fathers who did not use the Info library (p=0.0002; Chi² value=22.4),“ - See remark 3 of the reviewer.
  32. Line 214: „Likewise, parents who installed the app in 2019 were more likely to have watched the educational 214 video (51%, Table 1).“ Number is not presented in table 1. „more likeley“ in comparison to what?
  33. Line 224: „It is striking that initial viewing of the video about fever (46% versus 36%) and use of the Info library (73% versus 21%) are strongly associated with intensity of documentation of fever and to the maximum level of the child's fever in the app (Figure 2),“ Is this really surprising? There are users and non-users. Therefore, people using the app are obviously interested in documentation and information both. However, the correlation with a higher maximum level of the child’s fever should be discussed further.
  34. Line 231: “manageable” is used two times.
  35. Some problems exist with the formatting of the references (e.g. reference 4 and 12).
